# Immunogenic Cell Death Photothermally Mediated by Erythrocyte Membrane-Coated Magnetofluorescent Nanocarriers Improves Survival in Sarcoma Model

**DOI:** 10.3390/pharmaceutics15030943

**Published:** 2023-03-14

**Authors:** Ailton Antonio Sousa-Junior, Francyelli Mello-Andrade, João Victor Ribeiro Rocha, Tácio Gonçalves Hayasaki, Juliana Santana de Curcio, Lívia do Carmo Silva, Ricardo Costa de Santana, Eliana Martins Lima, Cléver Gomes Cardoso, Elisângela de Paula Silveira-Lacerda, Sebastião Antonio Mendanha, Andris Figueiroa Bakuzis

**Affiliations:** 1Institute of Physics, Federal University of Goias, Goiania 74690-900, GO, Brazil; 2FarmaTec, School of Pharmacy, Federal University of Goias, Goiania 74690-631, GO, Brazil; 3Biological Sciences Institute, Federal University of Goias, Goiania 74045-155, GO, Brazil; 4Federal Institute of Education, Science and Technology of Goias, Goiania 74055-110, GO, Brazil; 5CNanoMed, Federal University of Goias, Goiania 74690-631, GO, Brazil

**Keywords:** thermal nanomedicine, cancer, near-infrared dye, biomimetic nanoparticles, S180 cells

## Abstract

Inducing immunogenic cell death (ICD) during cancer therapy is a major challenge that might significantly improve patient survival. The purpose of this study was to develop a theranostic nanocarrier, capable both of conveying a cytotoxic thermal dose when mediating photothermal therapy (PTT) after its intravenous delivery, and of consequently inducing ICD, improving survival. The nanocarrier consists of red blood cell membranes (RBCm) embedding the near-infrared dye IR-780 (IR) and camouflaging Mn-ferrite nanoparticles (RBCm-IR-Mn). The RBCm-IR-Mn nanocarriers were characterized by size, morphology, surface charge, magnetic, photophysical, and photothermal properties. Their photothermal conversion efficiency was found to be size- and concentration-dependent. Late apoptosis was observed as the cell death mechanism for PTT. Calreticulin and HMGB1 protein levels increased for in vitro PTT with temperature around 55 °C (ablative regime) but not for 44 °C (hyperthermia), suggesting ICD elicitation under ablation. RBCm-IR-Mn were then intravenously administered in sarcoma S180-bearing Swiss mice, and in vivo ablative PTT was performed five days later. Tumor volumes were monitored for the subsequent 120 days. RBCm-IR-Mn-mediated PTT promoted tumor regression in 11/12 animals, with an overall survival rate of 85% (11/13). Our results demonstrate that the RBCm-IR-Mn nanocarriers are great candidates for PTT-induced cancer immunotherapy.

## 1. Introduction

A new nanotechnology strategy, named cell membrane-coated nanoparticles (CMNPs), is under development and is expected to have a great impact in the clinic, for instance, in cancer therapy [1]. The reason is because CMNPs associate the biological functions of membranes (immune evasion, immune stimulation, cancer targeting, etc.) with organic/inorganic nanostructures to create theranostic biomimetic core–shell nanoparticles. This technology aims to improve drug delivery, photothermal therapy (PTT), immunotherapy (IMT), and others [1].

The CMNPs might contain imaging tracers for near-infrared, PET, and/or magnetic particle imaging (MPI), contrast agents for MRI, and magneto- and/or photoresponsive particles/molecules for thermal nanomedicine. Several membranes extracted from different cells have been used; the primer reports include erythrocytes, platelets, macrophages, stem cells, cancer, and bacteria [1,2,3]. However, one of the most popular CMNPs uses erythrocyte membranes as coating material, maybe due to its potential for clinical translation. Coating nanoparticles with red blood cell membranes (RBCm) enhanced the blood circulation time and increased the intratumoral drug delivery [4,5]. Likewise, several RBCm-coated inorganic nanostructures have been investigated as CuS, Au, silica, and iron oxide NPs [1]. Among them, iron oxide-based NPs are of great interest since several products with this type of material have already been approved in the clinic for applications as MRI contrast agents for the treatment of anemia and magnetic hyperthermia [6].

Antonelli et al. were probably one of the first to investigate the interaction of iron oxide nanoparticles with erythrocyte membranes aiming at MRI and MPI imaging applications [7,8,9]. However, in those studies, the nanoparticles are internalized by red blood cells (RBC) and not surface-coated with the cell membrane. A few years later, Ren et al. successfully surface-coated magnetite nanoparticles with RBCm, demonstrating longer blood circulation time, higher intratumoral delivery efficiency, and MRI and PTT applications [10]. Similar applications were reported by Rao et al. using a microfluidic electroporation strategy to surface-coat magnetic nanoparticles [11]. More recently, our group developed Mn-ferrite nanoparticles surface-coated with RBCm embedding IR-780 dyes (RBCm-IR-Mn) [12]. We demonstrated a near-infrared imaging (NIR) application, which allowed for the investigation of the pharmacokinetics and intratumoral delivery efficiency of the nanocarrier. Moreover, a theoretical model was proposed allowing for the determination of both the maximum nanoparticle uptake and optimal delivery time. The study has interesting PTT applications since it establishes a clinical protocol for the therapy, correlating pharmacokinetic parameters with intratumoral delivery [12].

IR-780 is a multifunctional heptamethine dye with applications in NIR imaging, chemotherapy (CMT), photodynamic therapy (PDT), and PTT [13,14]. The molecule has better imaging and therapeutic properties than the clinically approved ICG dye [13], while its lipophilicity characteristic allows for easy membrane incorporation. Nevertheless, like several dyes, it might be limited by the photobleaching phenomenon. Furthermore, Mn-doped iron oxide nanoparticles were chosen because of their excellent response to alternating magnetic fields at low field conditions aiming for magnetic hyperthermia clinical applications [15,16], nanoparticle real-time detection and imaging capability with the alternating current biosusceptometry (ACB) technique [17,18,19], and potential PTT applications [14].

The purpose of this study is to evaluate in detail the in vitro and in vivo performance of RBCm-IR-Mn nanocarriers for photothermal therapy applications in the ablation regime using the murine sarcoma model S180 in Swiss albino mice. Several characterization techniques are used to investigate the biomedical potential of this nanocarrier. We compare the core and shell constituents’ properties in a systematic way. For instance, the photothermal conversion efficiency (PCE) is determined using Roper’s method [20]. A comparison of PCE between IR-780 incorporated in RBC vesicles (RBCm-IR), free Mn-ferrite nanoparticles (Mn), Mn-ferrite nanoparticles associated to RBC vesicles (RBCm-Mn), and our multifunctional nanocarrier containing all those components (RBCm-IR-Mn) is presented. This type of study is very rare, since the PTT efficiency of Mn-ferrite inorganic NPs has not been investigated, nor that of RBCm-Mn or RBCm-IR-Mn nanocarriers.

Furthermore, we also evaluated how the tumor heterogeneity influences the thermal dose delivery at constant laser power condition in the ablation regime, as well as if the intratumoral heat generation during PTT is influenced by the tumor matrix. The role of animal gender on the heat delivery was also addressed. Finally, the tumor growth profiles after PTT mediated by systemic administration of RBCm-IR-Mn nanocarriers were monitored up to 120 days to evaluate the possibility of tumor recurrence. We present strong evidence that PTT, at the ablative regime, induced an immunological response, resulting in a very high survival rate—85%—in the murine S180 model. The phenomenon is probably due to a combination of factors—high thermal dose, near-infrared chemotherapeutic action, and the adjuvant immune role of the metal-based nanoparticle—but one can conclude that the incorporation of the inorganic magnetic NPs in RBCm-IR-Mn nanocarriers enhances the PTT applications.

## 2. Materials and Methods

### 2.1. Chemicals

IR-780 iodide and methylamine (CH_3_NH_2_) were purchased from Sigma-Aldrich (St. Louis, MO, USA). Manganese (II) chloride tetrahydrate (MnCl_2_.4H_2_O), iron (III) chloride he-xahydrate (FeCl_3_.6H_2_O), and iron (III) nitrate nonahydrate (Fe(NO_3_)_3_.9H_2_O) were pur-chased from Vetec (Rio de Janeiro, Brazil). Nitric acid (HNO_3_), acetone, and sodium citrate tribasic dihydrate (Na_3_C_6_H_5_O_7_.2H_2_O) were purchased from Cromoline (Diadema, Brazil). Hydrochloric acid (HCl) was purchased from Qhemis (Jundiaí, Brazil). All other reagents were of analytical grade, and buffer solutions were prepared with Milli-Q water.

### 2.2. Synthesis of Mn-Ferrite Nanoparticles

The coprecipitation method was used to synthesize the MnFe_2_O_4_ magnetic NPs. First, we prepared precursor solutions. MnCl_2_.4H_2_O (197.91 g/mol) in solution (0.5 M, 98.95 g dissolved in 50 mL of HCl and 950 mL of H_2_O) provided the Mn^+2^ ions. FeCl_3_.6H_2_O (270.30 g/mol) in solution (1 M, 270.30 g dissolved in 50 mL of HCl and 950 mL of H_2_O) provided the Fe^+3^ ions. Next, we diluted 90 mL of methylamine in a beaker containing 400 mL of distilled water. We then applied magnetic stirring and heated this solution until boiling. After, we poured in 50 mL of each of the precursor solutions. Once the mixture boiled again, we conducted 30 min of controlled magnetic stirring and heating. Meanwhile, a black precipitate gradually formed. After 30 min, we magnetically separated the black precipitate containing the MnFe_2_O_4_ magnetic NPs and washed it three times with distilled water.

To protect the NPs’ surface against oxidation and corrosion, we submitted NPs to a process called passivation. For this purpose, we previously prepared aqueous solutions of HNO_3_ (63.01 g/mol) and (Fe(NO_3_)_3_.9H_2_O (404.00 g/mol), both at 0.5 M. Then, we poured 50 mL of the HNO_3_ solution on the washed magnetic NPs from the last step. Next, we poured in 50 mL of the (Fe(NO_3_)_3_.9H_2_O solution, applied magnetic stirring, and heated this dispersion until boiling. Once the dispersion started boiling, we conducted 30 min of controlled magnetic stirring and heating. After 30 min, we let the colloid cool down at room temperature. We then magnetically separated the passivated NPs, discarded the supernatant, and washed the passivated NPs three times with acetone. As soon as the acetone completely evaporated, we resuspended the passivated NPs in distilled water. 

The final step consisted of coating the now-passivated NPs with citrate ions. The goal of this step was to produce a stable colloid, with NPs less subject to agglomeration (due to magnetic attraction), and consequently, less subject to precipitation. Therefore, to the dispersion containing the passivated NPs, we added 1 mol of sodium citrate (Na_3_C_6_H_5_O_7_.2H_2_O, 294.10 g/mol) for every 10 mol of iron (Fe, 56 g/mol) in the dispersion. The dispersion became turbid and brownish. We then applied magnetic stirring and heating, keeping the dispersion at 80 °C for 10 min. After 10 min, we let the colloid cool down at room temperature. We then magnetically separated the coated NPs, discarded the supernatant, and washed the coated NPs three times with acetone. As soon as the acetone completely evaporated, we resuspended the citrate-coated MnFe_2_O_4_ NPs in distilled water, finishing the magnetic fluid preparation.

### 2.3. RBCm-IR-Mn Nanocarriers Preparation 

Red blood cell (RBC) vesicles were prepared from intact erythrocytes that were submitted to hypotonic lysis to originate hemoglobin-free membranes, following a previously described protocol [12]. Briefly, human blood, obtained from blood banks, was diluted in a phosphate saline buffer (PBS, 10 mM phosphate, 154 mM NaCl, pH = 7.4) and centrifuged at 150× *g* during 10 min at 4 °C to isolate the RBC. Then, the plasma and white blood cells were carefully removed by aspiration in three different cycles. To isolate the erythrocyte membranes, isolated RBCs were diluted in a lysis buffer solution (5 mM of phosphate, pH = 8.0) for 12 h at 4 °C and subsequently centrifuged at 25.000× *g* for 10 min at 4 °C. To remove the residual hemoglobin molecules, a second incubation in the lysis buffer solution was performed, followed by five consecutive centrifugations at 25.000× *g* for 10 min at 4 °C. At the end of this process, erythrocyte membranes with a whitish color were obtained. 

After RBC inner content removal and membrane isolation, the citrate-coated MnFe_2_O_4_ magnetic NPs’ dispersion (≈4.0 mg/mL in PBS), previously filtered with a Millex-GN syringe filter (220 nm pore), was added to the RBC-membranes (RBCm) suspension (at ≈2.0 mg of protein/mL), and the final solution was rigorously extruded using polycarbonate filters with 0.2 µm pores to form RBCm-Mn vesicles. IR-780 was incorporated into RBCm vesicles by a postinsertion protocol. A stock solution containing the dye diluted in chloroform was used to prepare IR-780 thin films in the bottom of glass tubes under a gaseous nitrogen flow. The RBCm nanocarriers were placed in contact with IR-780 films under slow orbital shaking (IKA, KS 400) for 30 min, and the excess of dye and NPs was removed by filtration.

### 2.4. RBCm-IR-Mn Nanocarriers Characterization

Biophysical characterization of RBCm-IR-Mn nanocarriers was performed, and their morphology, size, concentration, zeta potential, protein, magnetic NP, and IR-780 content were determined. RBCm vesicles’ morphology was assessed by transmission electron microscopy (TEM) using a JEOL JEM-2100 microscopy (Tokyo, JP). The samples were fixed using buffered formaldehyde solution (25%, pH = 7.0) and postfixed with osmium tetroxide 4% solution. After that, they were dehydrated in ethanol, deposited on the carbon film of a TEM copper grid, and finally colored by 0.5% aqueous uranyl acetate. 

Nanocarriers’ size and concentration were obtained by nanoparticle-tracking analysis (NTA, NanoSight NS500 equipped with a 532 nm laser and an EMCCD 215S camera, NanoSight, Amesbury, UK). RBCm suspensions were diluted in a ratio of 1:10^5^ before analysis and automatically injected into the sample compartment. Final samples concentrations and size distributions were obtained using the NTA 3.4 software. Zeta-potential measurements were performed using a Zetasizer Lab (Malvern Panalytical, Westborough, MA, USA). 

The total protein content of RBCm samples was determined using a commercial kit (Sigma-Aldrich, St. Louis, MO, USA) based on the reaction of bicinchoninic acid (BCA). 

The magnetic properties of the RBCm-IR-Mn nanocarriers were obtained by vibrating sample magnetometry (VSM), using a EV9 magnetometer (ADE Magnetics, USA). The saturation magnetization of each RBCm-IR-Mn sample was compared with the saturation magnetization of the free magnetic NPs (obtained from a dried powder sample) and with the erythrocyte membrane diamagnetic contribution, to determine their final magnetic concentration. 

Subsequently, IR-780 content and absorption curves were obtained in a Cary 50 UV-Vis spectrophotometer equipped with a full spectrum Xe pulse lamp single source (Varian Inc., Palo Alto, CA, USA). While the IR-780 content was determined using a calibration curve prepared with known dye concentrations, the absorption curves were recorded within the 200–1000 nm wavelength range using diluted samples to avoid scattering effects. 

Finally, emission measurements were performed for liquid samples in a quartz cuvette with 1 mm of optical length at room temperature using a Horiba Jobin Yvon spectrofluorimeter, Model Fluorolog-3 (FL3-221), equipped with an external laser source of 804 nm (80 mW), connected to an Spectracq2 data acquisition module and a R5509-73 PMT InGaAs detector for IR measurements.

### 2.5. Murine Sarcoma 180 Tumor Model

Swiss albino mice 6–8 weeks old with an average body weight of 25–35 g were used for tests. The animals were maintained under standard laboratory conditions (22–25 °C with dark/light cycle 12/12 h) with free access to a standard dry pellet diet and water ad libitum.

Murine Sarcoma 180 (S180) tumor cells were obtained from the Cytogenetic Laboratory, Genetic and Biochemistry Institute, Federal University of Uberlândia (Uberlândia, Minas Gerais, Brazil). The cancer cells were introduced intraperitoneally as ascites tumors into Swiss mice for 7 days and then used for both in vitro and in vivo assays. 

For in vivo tests, cell dilutions with 90% of viable cells were then used to induce solid tumors in mice via subcutaneous injection of 1 × 10^7^ cells in the dorsal region. Tumor volume measurements were performed with a digital caliper. Tumor volume over time was calculated as follows:(1)$$V\left[{\mathrm{mm}}^{3}\right]\approx \frac{Dd^{2}}{2}$$
where D is the long axis and d is the short axis of the solid tumor in millimeters (the implicit assumption being that the tumors are ellipsoidal).

### 2.6. Cell Culture and Viability Assays

S180 cells were obtained from S180-bearing Swiss mice, and then cultured in suspension in RPMI 1640 medium (Sigma, St. Louis, MO, USA), supplemented with 10% of FBS (fetal bovine serum) (Gibco^®^, Life Technologies, Carlsbad, CA, USA) and 1% of penicillin-streptomycin, in a humidified incubator (Thermo Fisher Scientific, Waltham, MA, USA) for 24 h at 37 °C with 10% CO_2_. Subsequently, cells were exposed to different concentrations of IR or RBCm-IR-Mn (3.9 to 500 μg/mL) and RBCm or Mn (19.5 to 2500 μg/mL) for 24 h. Posteriorly, the cell viability was measured by MTT assay (3-(4,5-dimethylthiazol-2-yl)-2,5-diphenyltetrazolium bromide) as previously described [14].

### 2.7. Assessment of Apoptosis by Annexin V-FITC/PI Staining

S180 cells were treated with RBCm-IR-Mn at 500 µg/mL. After 24 h, S180 cells were submitted to PTT using a laser power of 965 mW. Apoptosis was measured using the annexin V-fluorescein isothiocyanate (FITC)/propidium iodide (PI) double staining method according to the manufacturer’s instructions (BD Biosciences, San Jose, CA, USA). Flow cytometric analysis was performed immediately after annexin V-FITC/PI staining. Cells were analyzed by flow cytometry (FACSCalibur, BD Biosciences). The positive criteria for cells in early apoptosis were both annexin V-positivity and PI-negativity, whereas for cells in late apoptosis the positive criteria were both annexin V-positivity and PI-positivity.

### 2.8. Measurement of HMGB1 and Calreticulin (CRT) Expression

After PTT treatment, S180 cells were washed twice with 1× PBS and fixed with 4% paraformaldehyde (PFA) for 5 min. Cells were centrifuged at 1500 rpm for 3 min and washed again with PBS. 

To measure intracellular HMGB1, S180 cells were incubated for 15 min with permeabilization buffer and washed with PBS (containing 0.1% BSA and 0.1% Triton-X). Then, cells were incubated in the dark at room temperature for 1 h with anti-HMGB1 primary antibody (Abcam, Cambridge, MA, USA) at 1:10^3^. S180 cells were then washed and stained with anti-mouse IgG-FITC (Sigma-Aldrich) for 30 min. Finally, cells were washed twice with PBS, centrifuged for 5 min at 1500 rpm, resuspended in PBS, and analyzed by flow cytometry (FACSCanto II, BD Biosciences). 

To measure CRT externalization, S180 cells containing RBCm-IR-Mn were incubated for 1 h with anti-CRT antibody (Abcam Cambridge, MA) at 1:500 in 0.1% BSA/PBS. Next, cells were washed and stained with a secondary antibody (anti-rabbit IgG-FITC, Sigma-Aldrich) for 30 min. Samples were washed, centrifuged, resuspended in PBS, and analyzed by flow cytometry (FACSCanto II, BD Biosciences).

### 2.9. Photothermal Experiments

Photothermal therapy (PTT) experiments used a diode laser, model iZi 808, bought from LASERLine (São Paulo, Brazil), with 808 nm wavelength. The sample holder contained 500 µL of the nanocarrier dispersion. The photothermal properties of the nanocarrier were investigated at 965 mW. The temperature was monitored using an infrared thermal camera bought from FLIR, model SC620 (Wilsonville, OR, USA). A region of interest (ROI) centered at the laser spot on the sample was used to report the mean temperature during PTT.

Four PTT studies were performed. To obtain the photothermal conversion efficiency (PCE), temperature was monitored for 90 min (laser on) and cooling was monitored for 10 min (laser off), except when the sample showed photobleaching. For the photostability experiments, laser was on for 5 min, and three PTT sections were performed sequentially. Finally, for the in vitro and in vivo studies, laser was on for 15 min, and cooling was not monitored. 

Animals with tumors within 50–150 mm^3^ were considered for the in vivo PTT study. Four groups were investigated. The control group consisted of females with S180 and no PTT (n=4), while three other groups evaluated PTT treatments, namely females with S180 (n=4), group GF; males with S180 (n=4), group GM; and females with S180 that previously received losartan (n=5), group GFL. Day-5 was established as the day of the intravenous administration of the nanocarriers. The process consisted of retro-orbitally injecting 150 μL of the RBCm-IR-Mn dispersion, as discussed previously in the PK study [12]. Five days after administration, the animals were treated with PTT (Day 0).

During the PTT procedures, the animals were anesthetized by approximately 60 µL (equivalent to 2 µL/g of body weight, for an average 30vg mouse) of a solution containing ketamine (87.5 mg/mL) and xylazine (2.5 mg/mL). The laser power was fixed at its highest value, 965 mW, and the surface temperature of the tumor was monitored with the thermal camera. The angle of the camera was perpendicular to the skin surface to avoid errors on temperature measurements since the tumor has a spherical shape [21]. We established the threshold limit of 55 °C during PTT. Therefore, if during PTT this temperature was achieved, the laser power was tuned to control heat delivery. Almost all animals were treated by only one PTT therapeutic procedure; the only exception is discussed in the following.

### 2.10. Thermal Dose Determination

In the literature, CEM43 (cumulative equivalent minutes at 43 °C) is used for thermal dose determination. In our study, we focus on the ablative regime (temperatures higher than 50 °C). In this case, following the discussion by Pearce [22], the thermal dose at ablative conditions, for instance, at a temperature of 50 °C, CEM50, is determined from
(2)$$\mathrm{CEM}50\;\left[\min\right]=\sum_{t=t_{0}}^{t_{\mathrm{final}}}{\left[R_{\mathrm{CEM}50}\right]}^{50-\bar{T}}\Delta t$$
where $\bar{T}$ represents the mean temperature within the interval Δt; $t_{0}$ and $t_{\mathrm{final}}$ correspond, respectively, to the initial and final times of PTT; while $R_{\mathrm{CEM}50}$ is a constant that quantifies the cell death rate at the reference temperature. $R_{\mathrm{CEM}50}=0.51$ for $\bar{T}>50$ °C, and $R_{\mathrm{CEM}50}=0.27$ for $\bar{T}\leq 50$ °C. The CEM50 was correlated to the CEM43, assuming that the activation energy was temperature-independent and noting that $R_{\mathrm{CEM}50}={\left(R_{\mathrm{CEM}43}\right)}^{0.9575}$ [22].

### 2.11. Photothermal Conversion Efficiency Determination

The photothermal conversion efficiency (PCE) was obtained using Roper’s model for the cooling process [20]. PCE measures the percentage of the nonionizing laser irradiation that is converted into heat. Basically, the PCE value η is obtained from the equation
(3)$$\eta =\frac{hS\left(T_{\max}-T_{\mathrm{env}}\right)-\dot{Q_{0}}}{P\left(1-10^{-A_{\lambda }}\right)}$$
where $T_{\max}$ and $T_{\mathrm{env}}$ are the stationary (maximum) and environment temperatures, respectively; $\dot{Q_{0}}$ is the nonspecific power absorbed by the sample holder and liquid carrier; P is the laser power; $A_{\lambda }$ is the sample absorbance at the wavelength λ; h is a heat transfer coefficient; and S is the surface area irradiated during PTT. All those terms can be obtained experimentally, as discussed by Roper [20], except for hS, which is determined after the analysis of the cooling profile (for details, please check the Appendix A of ref. [23]). Briefly, defining a dimensionless temperature as
(4)$$\Theta \left(t\right)=\frac{T_{\mathrm{env}}-T\left(t\right)}{T_{\mathrm{env}}-T_{\max}}$$
and solving the power balance equation for a sample irradiated with a laser power P and wavelength λ, it is possible to show that the solution during the cooling is
(5)$$\Theta \left(t\right)=e^{-t/\tau_{s}}$$
where t is the time elapsed after turning off the laser, while $\tau_{s}$ is a thermal relaxation time constant given by
(6)$$\tau_{s}=\frac{\sum_{i}m_{i}c_{p,i}}{hS}$$

Note that the sum is over all the materials involved, i.e., the sample holder, the carrier liquid, and the nanocarrier. $m_{i}$ and $c_{p,i}$ are the mass and specific heat of the material i, respectively. In summary, by fitting $\Theta \left(t\right)$ (Equation (5)) to the cooling data, it is possible to determine $\tau_{s}$, and subsequently, the product hS (Equation (6)). Knowing hS and other parameters obtained experimentally, we can calculate the PCE (η, Equation (3)).

### 2.12. Statistical Analysis

Statistical analyses were performed via analysis of variance (ANOVA) and/or student’s *t*-test using OriginPro^®^ and GraphPad Prism. Statistically significant differences were considered for p<0.05.

## 3. Results

### 3.1. RBCm-IR-Mn Nanocarrier Characterization and Photothermal Properties

Figure 1 summarizes the biophysical properties of the RBCm-IR-Mn nanocarriers used in this work. As can be seen in Figure 1a, RBCm-IR presented a morphology compatible with a cell membrane vesicle. Figure 1b shows the actual RBCm-IR-Mn nanocarriers, with similar morphology, despite the additional Mn-ferrite NPs cargo. Aggregates of Mn-ferrite NPs can also be seen in their surroundings. Additional TEM images and EDS spectra can be found in Appendix A, as well as in the Appendix A of our previous study [12]. 

Figure 1c shows the mean size obtained by nanoparticle tracking analysis (NTA) and the zeta potential values for RBCm-IR-Mn, RBCm-IR, and RBCm-Mn, as well as for Mn NP aggregates. While RBCm-Mn and RBCm-IR-Mn presented sizes of 113 ± 71 nm and 143 ± 79 nm, respectively, the diameter observed for RBCm-IR was slightly higher, at 220 ± 103 nm. On the other hand, the Mn NP aggregates’ mean size was found to be 91 ± 56 nm. Nevertheless, RBCm-IR-Mn sizes were around 100 nm, which is traditionally preferable for in vivo applications. NTA size distributions for these samples are provided in Appendix A.

Regarding the zeta potential (Figure 1c), the mean values range from −20 to −26 mV, which indicates the colloidal stability of RBCm-IR, RBCm-Mn, and RBCm-IR-Mn, being in agreement with values previously reported by our group [12], as well as with other studies in the literature (see ref. [10] and references therein). 

The magnetic responses of RBCm-Mn and RBCm-IR-Mn nanocarriers were obtained by vibrating sample magnetometry (VSM) and are shown in Figure 1d. As one can see, both RBCm vesicles containing magnetic nanoparticles exhibited similar saturation magnetization, indicating that both RBCm-Mn and RBCm-IR-Mn have approximately the same Mn NP cargo. This result also corroborates with the success of Mn NPs’ incorporation into the RBC membranes.

Figure 1e shows the absorption spectra of RBCm-IR, RBCm-Mn, and RBCm-IR-Mn, as well as those of IR-780 dispersed in ethanol and of Mn-ferrite NPs. Maximum absorption is observed for free IR-780 at 784 nm, followed by the RBCm-IR sample at 800 nm. An absorption peak can also be observed at 800 nm for RBCm-IR-Mn, but its absorption spectrum is strongly dominated by the Mn NPs. 

Figure 1f shows the photoluminescence of RBCm-IR, RBCm-Mn, and RBCm-IR-Mn, excited at 804 nm. As expected, the RBCm-Mn did not present any emission due to the absence of IR-780 molecules in its composition. In contrast, RBCm-IR and RBCm-IR-Mn presented two characteristic peaks. Emission peaks were observed at 834 and 932 nm for RBCm-IR, while for RBCm-IR-Mn, we verified a red-shift on the first peak (to 839 nm), and the second peak remained at 932 nm. Quenching was also observed at RBCm-IR-Mn’s first peak, probably due the presence of Mn NPs. Once again, one can see an indication of both IR-780 and Mn-ferrite NPs’ successful incorporation.

Next, we investigated the photothermal properties of the nanocarriers at 965 mW. Figure 1g shows the temperature profile during PTT for RBCm-IR, RBCm-Mn, and RBCm-IR-Mn. At this selected experimental condition, RBCm-IR showed photobleaching indicating limitations of this nanocarrier for PTT applications. The highest temperature was achieved by RBCm-Mn, but ablation temperatures were also observed for RBCm-IR-Mn, suggesting that both nanocarriers have potential for PTT applications. The inset shows the photostability assay for RBCm-IR-Mn, while Appendix A shows the photostability results for RBCm-IR. Clearly, RBCm-IR is not stable for PTT, while RBCm-IR-Mn is not affected by at least three heating cycles. This result indicates that the incorporation of inorganic nanoparticles in the RBCm vesicles enhances the PTT properties of the nanocarrier. 

To obtain the photothermal conversion efficiencies (PCE), we evaluated the samples’ cooling regimes using Roper´s method (Figure 1h). RBCm-IR PCE was not evaluated as it presented photobleaching. The inset shows the cooling regime (the first three minutes after turning off the laser) for the RBCm-Mn sample. The solid line represents the best fit of Roper’s model (for the cooling regime) to the experimental data. Appendix A shows the result for the RBCm-IR-Mn sample. The analysis revealed a thermal relaxation time of 248 s for RBCm-IR-Mn and 237 s for RBCm-Mn. Using Equation (3), we calculated the PCE values for both samples, and we found 17% for RBCm-IR-Mn and 30% for RBCm-Mn. We also calculated the PCE for Mn NPs, and it was found to be 34% (Figure 1h).

### 3.2. IR-780 Near-Infrared Dye Shows Chemotherapeutic Action

Cell viability studies of RBCm-IR-Mn without laser irradiation were performed. Appendix A presents the MTT study of free IR-780 as a function of concentration. We found that IR-780 has a chemotherapeutic effect even without laser irradiation, with an IC50 of 36.7 ± 4.5 µg/mL. Appendix A shows the cell viability investigation of the RBCm vesicles as a function of concentration. The data demonstrate, as expected, that RBCm vesicles are biocompatible. Appendix A shows the MTT study of the Mn-ferrite nanoparticles. No toxic effect was found, even by increasing the concentration of NPs up to 2500 µg/mL. Again, there is no toxicity arising from the magnetic nanoparticles at this concentration range. Appendix A shows the MTT study of RBCm-IR-Mn. Increasing the RBCm-IR-Mn concentration resulted in a decrease in S180 cells’ viability. At the highest dose tested, 500 µg/mL, the viability remained around 65%. The decrease in the viability arises from the IR-780 dye incorporated in the RBCm. One can conclude that the toxicity of the nanocarrier is not high.

### 3.3. Photothermal Therapy Induces Immunogenic Cell Death

Figure 2a shows the in vitro photothermal experiments with S180 cells irradiated with a laser power of 965 mW. The data show the mean temperature profile of the control group (S180 cells without NPs) and of groups with S180 cells incubated with RBCm-IR-Mn at 500 µg/mL and at 1000 µg/mL. The maximum temperature achieved during PTT for the control group was 31 °C, while when incubated with the nanocarriers it was 44 °C and 54 °C, respectively, for the lower- and higher-concentration groups. As expected, the thermal dose was concentration-dependent. 

Figure 2b shows the analysis of flow cytometry data for several experimental conditions, with and without PTT, with RBCm-IR-Mn at 500 µg/mL. It shows the percentage of viable cells for distinct cell death mechanisms (early apoptosis, late apoptosis, and necrosis). Without PTT, the presence of RBCm-IR-Mn resulted in slight differences between early and late apoptosis, but there was no statistical difference between the groups. On the contrary, with PTT, the percentage of viable cells changed from 92 ± 2% for the control to 72 ± 7% for the nanocarrier group. Meanwhile, the percentage of early apoptosis (necrosis) changed from 4 ± 2% (5 ± 2%) for the control to 2 ± 1% (0%) for the nanocarrier group. A higher modification was found for late apoptosis: it changed from 5 ± 3% to 20 ± 4% for the nanocarrier group. These results clearly indicate that the late apoptosis mechanism is an important cell death mechanism for PTT.

Figure 2c shows the cell viability study for RBCm-IR-Mn at 500 µg/mL, with and without laser irradiation. The cell viability decreased from 82 ± 2% without to 34 ± 1% with PTT. Figure 2d investigates the intracellular HMGB1 expression 24 h after PTT at two distinct nanocarrier concentrations. Results were obtained using flow cytometry, where a total of 10^4^ events were evaluated. We found evidence for immunogenic cell death (ICD) for the higher concentration case (1000 µg/mL), which is the one with the higher thermal dose (Figure 2a). The conclusion is corroborated by the evaluation of calreticulin (CRT), another ICD marker (Figure 2e). The results indicate a 2.7-fold increase in CRT compared to the control for the sample upon treatment with RBCm-IR-Mn at 1000 µg/mL and PTT.

### 3.4. Photothermal Therapy Induces Tumor Regression in Sarcoma 180-Bearing Mice

Preliminary tests irradiating tumors with the laser power of 965 mW induced nonspecific heating without achieving a significant temperature increase. Appendix A shows the temperature profile of one of these tests. The maximum temperature is close to 37.5 °C, corresponding to a temperature variation (relative to ambient temperature) of just 5 °C.

Figure 3a shows the experimental design of the in vivo PTT study with S180-bearing mice. Mice in the control group did not receive RBCm-IR-Mn, and their tumors were not irradiated by laser. Three other groups were evaluated, namely GF (females, n=4), GFL (females treated with losartan, n=5), and GM (males, n=5). Day-5 was defined as the day of the intravenous administration of RBCm-IR-Mn. On Day 0, the animals were sub-mitted to PTT at high laser power (ablation regime). The S180 tumors’ evolution was monitored as a function of time. Animals were followed for up to 120 days.

Figure 3b shows the PTT profiles of distinct animals of groups GF, GFL, and GM, respectively. One clearly observes that distinct thermal doses are observed, as expected, due to tumor heterogeneity and distinct intratumoral nanocarrier concentrations. For some animals, the laser power was tuned (decreased) to avoid temperatures higher than 55 °C. This is particularly clear for animal GFL-2E, but also happened for GM-SM.

Figure 3c shows the tumor evolution after PTT for distinct animals of groups GF, GFL, and GM, respectively. Almost all the animals showed complete tumor regression. Animals were monitored for up to 120 days to investigate the possibility of tumor recurrence. One animal, GF-2E, died for unknown reasons, while animal GM-SM showed a decrease in the rate of tumor growth in comparison to the control group, but no regression. The tumor growth kinetic of this animal is not shown in Figure 3c, but is discussed in the next subsection.

Histopathological analysis of animal GM-1E, which showed complete regression, was performed 120 days after PTT. We found no tissue alteration related either to the nanocarrier administration or to the PTT treatment. Representative slides of liver, spleen, kidney, and lung tissues of animal GM-1E are shown in Figure 4a. There were no alterations in the tissue parenchyma, as well as no sign of cellular infiltration, hyperemia, necrosis, or apoptosis. No tumor tissue was found 120 days after treatment, and only connective and adipose tissue (dermis and hypodermis) were found in the histopathological analysis within the region where the tumor was implanted before treatment. Perls staining indicated that there was no retention of a considerable amount of iron (Fe) in tissues and in the tumor region after 120 days.

### 3.5. Animal GM-SM Did Not Respond to Additional PTT Sessions

Appendix A shows the tumor growth kinetics of animal GM-SM. Although the growth rate is slower than that of the control group, the tumor kept growing. To check if other PTT sessions could manage tumor growth, two other PTT procedures were performed. 

Appendix A shows the PTT temperature profiles for animal GM-SM at distinct days. T1 corresponds to PTT 5 days after RBCm-IR-Mn administration, T2, 35 days, and T3, 44 days. Those points are also shown in Appendix A as a function of the time after the first PTT treatment. Since we were not able to control tumor growth, the animal was sacrificed on day 45 after T1. 

Histopathological analysis of GM-SM can be found in Figure 4b, which shows the HE and Perls staining of several organs. There is no histopathological alteration in the liver, spleen, kidney, and lung tissues. Furthermore, Perls staining showed moderate presence of iron in the spleen and an intense amount in the tumor tissue, indicating a considerable amount of MnFe_2_O_4_ NPs in the tumor up to 45 days after PTT due to efficient nanocarrier intratumoral delivery.

### 3.6. RBCm-IR-Mn-Mediated PTT Resulted in High Survival Rate

Figure 5a shows the thermal dose in terms of CEM50 for animals of all three groups, GF, GFL and GM. Error bars represent standard deviation. The mean CEM50 of GF is 157 min, and for GFL is 119 min, while for GM it is 106 min. According to statistical analysis, there is no significant difference between the groups. For this reason, all animals were included for Kaplan–Meyer analysis.

Figure 5b shows the survival curves for the control and the PTT group. We found an outstanding survival rate of 85% after monitoring the animals for 120 days.

## 4. Discussion

Cell membrane-coated nanoparticles (CMNPs) generally consist of synthetic cores camouflaged by a layer of natural cell membranes capable of mimicking the original cell membrane properties [1]. Of all these properties, the ability to interact with biological substrates in the bloodstream while avoiding clearance along the path to the targeted organ is of particular interest. Indeed, this dual ability enables higher delivery efficiency of the nanoparticle cargo while minimizing off-target effects. In a previous study, we demonstrated that nanoparticles consisting of red blood cell membranes (RBCm) coating manganese (Mn) ferrite magnetic nanoparticles and embedding the near-infrared fluorescent dye IR-780, namely RBCm-IR-Mn nanocarriers, are more efficiently delivered to tumors than free IR-780 after intravenous injection [12]. RBCm-IR-Mn delivery efficiency was about two times higher than that of free IR-780 for Ehrlich tumors, and our pharmacokinetic model accurately predicted a delivery efficiency even higher (nearly 3.5-fold) for animals treated with losartan prior to RBCm-IR-Mn intravenous administration [12]. Building on our earlier findings, we herein investigate RBCm-IR-Mn’s potential as mediators of photothermal therapy (PTT).

In effect, CMNPs have been explored for their potential in PTT, utilizing either near-infrared dyes or inorganic nanoparticles such as gold, copper sulfide, and iron oxide NPs [1,5,24,25,26]. Among the iron oxide NPs, magnetite is the most often used, while ICG is the primary choice for dye-based PTT [1,27]. Both have been approved for clinical use [6]. Bahmani et al. were probably one of the first to incorporate ICG dyes in RBC membranes. The RBCm-based nanoparticles were investigated for NIR imaging and PTT [24], but the photobleaching phenomenon suggested limited applications. As for our RBCm-IR-Mn nanocarrier, IR-780 dye was chosen due to its superior photophysical properties compared to ICG [13], while Mn-doped iron oxide NPs were chosen for their theranostic applications, including low-field magnetic hyperthermia, PTT, MRI contrast agent applications, ACB magnetic tracers, and others [14,15,16,17,18,28]. Like Bahamani et al., we also observed photobleaching for the RBCm-IR (Figure 1g), indicating that this nanocarrier (a control sample in the context of this study) would have serious limitations for PTT. Nevertheless, our previous article demonstrated the near-infrared properties of the RBCm-IR-Mn nanocarrier at low laser power (80 mW), allowing for the noninvasive investigation of its pharmacokinetics and intratumoral delivery efficiency. Here, we thoroughly explore the in vitro and in vivo PTT applications of the RBCm-IR-Mn nanocarrier under the ablative regime.

Cryo-TEM is the gold standard technique employed to visualize organic nanocarriers. However, since access to cryo-TEM is quite limited, we combined conventional TEM and uranyl acetate staining to visualize our RBCm-IR-Mn nanocarriers. Figure 1a demonstrates that RBCm-IR samples exhibit an approximately spherical morphology, with sizes that are in agreement with the NTA results in Figure 1c (and Appendix A). Similarly, Figure 1b shows that RBCm-IR-Mn nanocarriers preserve this spherical morphology, despite the addition of the Mn-ferrite NPs. The TEM images and EDS spectra in Appendix A additionally demonstrate that the magnetic cargo can be found either enclosed by the RBCm or within the RBCm bilayer, as we have previously observed [12]. Furthermore, though uranyl acetate staining favors the imaging of the organic RBCm, it also impairs the imaging of the inorganic Mn-ferrite NPs, as uranium dominates the EDS spectra over Fe and Mn (Appendix A). Anyhow, the additional characterizations presented in this study, along with those from our previous work [12], confirm the successful assembly of RBCm-IR-Mn as designed.

It is noteworthy that although RBCm-IR seems to present a larger mean diameter (Figure 1c), there was no statistically significant difference found between the mean diameter of RBCm-IR and those of RBCm-Mn and RBCm-IR-Mn. Indeed, their size distributions are quite similar, with relatively higher polydispersity for RBCm-IR and RBCm-IR-Mn (Appendix A). This apparent difference in size is likely due to a difference in the number of serial extrusions performed to produce these samples: 10 extrusions for RBCm-IR and 20 for RBCm-Mn and RBCm-IR-Mn. A higher number of extrusions is needed when the sample contains Mn-ferrite NPs, since they form aggregates that hinder the process. Meanwhile, a higher number of extrusions for RBCm-IR would imply a greater loss in RBCm and, consequently, in IR-780. To be noted that RBCm-IR-Mn is the fully functional nanocarrier, whereas Mn-ferrite NPs, RBCm-IR and RBCm-Mn are primarily used as control samples for photophysical characterization. Therefore, only RBCm-IR-Mn was relevant for the in vivo experiments. Additional information relative to these control samples can be found in our previous study [12]. Moreover, considering the size distributions’ similarities and that no statistically significant difference was found between their mean values, no significant differences are expected for experiments involving these control samples.

Additionally, we observed that Mn-ferrite NPs dominate the absorption spectrum of RBCm-IR-Mn (Figure 1e), making it suitable for PTT applications, since unlike IR-780, Mn-ferrite NPs do not undergo photobleaching. In Figure 1e, absorbance values were primarily taken below 1.0. Indeed, we generally avoid measuring absorbances much greater than 1.0, as doing so would take us beyond the linearity domain of Beer–Lambert’s law, rendering it inapplicable. Therefore, although we prepared samples with nearly the same contents of IR-780, we had to dilute them using different dilution factors. As samples containing Mn-ferrite NPs required higher dilution factors, their dominance over IR-780 seems to be reinforced. On the other hand, in Figure 1f, the amount of IR-780 is approximately the same for both samples. The difference in photoluminescence intensity observed between RBCm-IR and RBCm-IR-Mn is no longer attributable to differences in dilution factors. Instead, it can be attributed to quenching phenomena resulting from the incorporation of Mn-ferrite NPs into the RBCm-IR sample, to produce RBCm-IR-Mn nanocarriers. Meanwhile, the red-shift observed for RBCm-IR relative to the free IR-780 sample (Figure 1e) accounts for the restriction in terms of degrees of freedom imposed to IR-780 as it is incorporated into the RBCm. After incorporation, some of its vibration modes are no longer allowed, resulting in the observed red-shift, and ultimately counting as evidence of the successful assembly of the RBCm-IR-Mn nanocarriers.

Gold-based nanostructures are the most popular materials for PTT due to their high PCE values, ranging from 30% to 100%, the higher values resulting from plasmonic resonance [29,30]. However, we are not aware of any inorganic Au-based nanoproduct already approved for clinical use, although some clinical studies are beginning [6]. In contrast, there are several iron oxide NPs approved in the clinic [6]. In our study, we showed that the Mn-doped iron oxide NP has a PCE value ranging from 25% to 47% for concentrations spanning from 1 to 30 mg/mL. These values are higher than those reported by Wang et al. for CuS NPs (16%), but within the range reported for magnetite NPs by Tian et al. (39%) [31,32]. In this study, we found PCE values of 30% and 17% for RBCm-Mn and RBCm-IR-Mn, respectively. The lower PCE value for RBCm-IR-Mn compared to RBCm-Mn is likely due to a lower content of magnetic nanoparticles in the RBCm vesicle (Figure 1d) and to scattering effects, since the mean size of RBCm-IR-Mn (143 nm) was relatively higher than that of RBCm-Mn (113 nm). The IR-780 dye is not playing any significant role in PTT at this experimental condition, but is obviously important for near-infrared imaging [12].

The S180 murine tumor model was used for the in vivo investigation using Swiss albino mice. The nanocarriers were systemically administered, and the in vivo PTT experiments were performed only 5 days later.

We evaluated how tumor heterogeneity influences thermal dose delivery at constant laser power in the ablation regime. Jain’s research group previously showed that losartan reduces collagen and hyaluronan production, affecting the tumor extracellular matrix, thus resulting in increased nanoparticle delivery [33]. Our previous study yielded the same conclusion [12]. Here, we evaluated whether the intratumoral heat generation during PTT is influenced by the tumor matrix. We investigated three groups: GFL (receiving losartan), GF (not receiving losartan), and GM (also not receiving losartan). Indeed, we found a higher mean CEM50 value for the GF group (157 min), compared to GFL (119 min) or GM (106 min). The lower thermal dose of GFL compared to GF might look in contradiction with the higher delivery efficiency observed for GFL [12]. However, losartan modifies the tumor matrix and possibly affects intratumoral blood vascularity. Therefore, it might be possible that the heat leak due to blood convection is responsible for the lower thermal dose in this group. Indeed, tumor heterogeneity between animals might explain, together with distinct nanocarrier concentrations, the different thermal doses between animals of all groups.

Another goal was to determine whether intravenous administration of RBCm-IR-Mn followed by PTT in the ablation regime could control tumor growth. To ensure proper evaluation of treatment efficacy, we monitored the animals for up to 120 days after PTT—a much longer timeframe than that reported in other studies, in which tumor recurrence could not be duly assessed due to much shorter monitoring periods [25,26].

Moreover, heat can induce immunological responses, but the ideal thermal dose, either for hyperthermia or for ablation, is still under debate. For instance, Toraya-Brown et al. found that magnetic hyperthermia at 43 °C induces a CD8+ T cell response, while no effect was observed for a reference temperature of 45 °C in a murine melanoma (B16F10) model [34]. On the other hand, Sweeney et al. observed immunogenic cell death (ICD) induced by PTT in the ablation range, for temperatures around 65 °C in a murine neuroblastoma (Neuro2a) model [35]. In our in vivo PTT study with the murine sarcoma S180 model, we established that the maximum tumor surface temperature should not exceed 55 °C. Therefore, if the mean temperature within the monitored ROI reached this threshold, we would adjust the laser power to maintain the treatment temperature around this reference value. We established the 55 °C temperature threshold based on the findings of Sweeney et al., who demonstrated that moderate ablative thermal doses could elicit ICD in their murine neuroblastoma model [35]. In our study, only a limited number of animals received treatment at the reference temperature of 55 °C for most of the PTT session duration (Figure 3b). One of the reasons for this outcome is that the nanocarrier concentration was lower than the optimum intratumoral level since PTT only occurred 5 days after nanocarrier intravenous administration. Nevertheless, the temperature ranges achieved were still in line with recently published results for in vivo PTT mediated by different inorganic nanoparticles surface-coated with cell membranes [25,26].

For instance, Yu et al. investigated magnetite NPs surface-coated with myeloid-derived suppressor cell membranes in the B16F10 model. The animals were monitored for 12 days after PTT. The authors observed ICD, with enhanced expression of HMGB1 and calreticulin (CRT) for PTT temperatures around 55 °C. Xiong et al. also investigated magnetite NPs surface-coated with ID8 ovarian cancer cell membranes fused with RBC membranes embedding ICG. Better PTT responses were observed for temperatures near 54 °C for the ID8 murine-derived ovarian cancer [25]. The animals were monitored for 16 days after PTT. Enhanced activation of CD4+ and CD8a+ T cells was reported, indicating a synergistic PTT immunotherapy. Similarly, our in vitro PTT investigation (Figure 2) demonstrated an enhanced expression of HMGB1 and CRT for temperatures close to 55 °C. This is strong evidence that ICD can be induced for S180 cells under this ablative regime. Furthermore, the late apoptosis observed upon PTT (Figure 2b) also corroborates ICD induction, since this cell death mechanism promotes the release of HMGB1 [36]. However, PTT and ICD alone might not be the only important factors.

Indeed, our previous study with bovine serum albumin (BSA) nanoparticles embedding IR-780 indicate that this near-infrared dye has a chemotherapeutic action even without laser irradiation [14]. This outcome corroborates the results in Appendix A. However, when incorporated in RBCm-IR-Mn, this chemotherapeutic effect is less pronounced due to the lower amount of IR-780 incorporated in the membrane (Appendix A). In addition, the generation of reactive oxygen species (ROS) mediated by IR-780 under high laser power irradiation cannot be neglected [37].

Another important factor could be related to the release of metal ions from the inorganic nanoparticle, which can happen through a biodegradation process. It is well known that the tumor microenvironment has an acidic pH, and that when internalized by cells, nanoparticles enter endosomes, which might trigger the biodegradation of inorganic materials due to their low pH. In our study, we use Mn-ferrite nanoparticles, so Mn^2+^ and Fe^3+^ are expected to be released with time. The iron content in the organs can be qualitatively evaluated by histopathology analysis. In Figure 4, we show the histopathology results for animals GM-1E (complete tumor regression) and GM-SM (which showed tumor recurrence). GM-1E was sacrificed 120 days after PTT, and GM-SM after 45 days. The Perls analysis for GM-SM shows evidence of a high amount of iron (probably due to the inorganic NPs) in the spleen and in the tumor, in contrast with animal GM-1E. The results suggest that within this time range, the inorganic nanoparticles might have been biodegraded. This corroborates a recent study of Miranda´s group, which demonstrated (using ACB), that citrate-coated Mn-ferrite nanoparticles are not only eliminated by the hepatic route (feces), but might also be biodegraded [17]. In addition, the inorganic nanoparticles are expected to promote an increase in the presentation of tumor antigens to immune cells (dendritic cells and macrophages) due to protein corona formation and to NP uptake in the tumor microenvironment.

Both Mn^+2^ and Fe^+3^ ions play important roles in many biological activities, including immune processes [38]. For instance, the iron content in magnetite nanoparticles is very important for the treatment of iron deficiency anemia. Several nanoproducts with this type of inorganic NPs have been approved for clinical use [6]. Like magnetite, Mn-ferrite nanoparticles have been established as interesting T2-weighted MRI contrast agents [28]. Furthermore, if biodegraded, interesting T1 applications might be activated [39,40]. Indeed, Mn^2+^ ions have great performance as T1-weighted MRI contrast agents and might be an alternative for Gd-based agents used in the clinic [38,39]. Additionally, it is becoming clear that not only Fe^+3^, but also Mn^+2^ ions have great importance on antitumor immune responses. There are reports relating Mn^+2^ ions to the activation of natural killer cells in mice, the release of cytokines, the polarization of M1 macrophages, and other functions [38]. Overall, the long-term biodegradation of inorganic nanoparticles and the release of their metal ions might also play a role in preventing tumor recurrence, hence allowing for a higher survival rate. This effect is enhanced when cell membrane-coated nanocarriers are utilized, due to their higher intratumoral delivery efficiency.

## 5. Conclusions

Theranostic nanocarriers, consisting of IR-780-stained red blood cell membranes (RBCm) camouflaging Mn-ferrite magnetic nanoparticles, named RBCm-IR-Mn, were successfully developed. Physicochemical characterizations revealed their spherical shape, mean hydrodynamic diameter within 100–150 nm, low polydispersity index, zeta potential around −25 mV, and superparamagnetic behavior under quasi-static magnetic field excitation, as well as absorption and fluorescence emission spectra, allowing for their use both in PTT and near-infrared imaging applications. Even though the embedded IR-780 undergoes photobleaching under ablative PTT, RBCm-IR-Mn photostability after multiple sequential PTT cycles is preserved by the photothermal converting Mn-doped iron oxide nanoparticles. Viability of murine sarcoma cells submitted to in vitro RBCm-IR-Mn-mediated PTT was significantly reduced after treatment, with late apoptosis standing out as the predominant cell death mechanism. Release of the typical immunogenic cell death (ICD) markers HMGB1 and calreticulin suggested the elicitation of antitumor immune responses in vitro. When intravenously administered into Sarcoma 180-bearing mice, RBCm-IR-Mn efficiently accumulated within the tumor microenvironment, enabling ablative PTT even 5 days after injection—3 days after the onset of systemic clearance, as per our previous pharmacokinetic study. In vivo PTT promoted rapid tumor regression in 11 out of 12 mice, with no relapses within 120 days after treatment, resulting in an overall survival rate of 85% (11/13). Together, our results demonstrate that RBCm-IR-Mns have great potential for PTT-induced cancer immunotherapy.

## Figures and Tables

**Figure 1 pharmaceutics-15-00943-f001:**
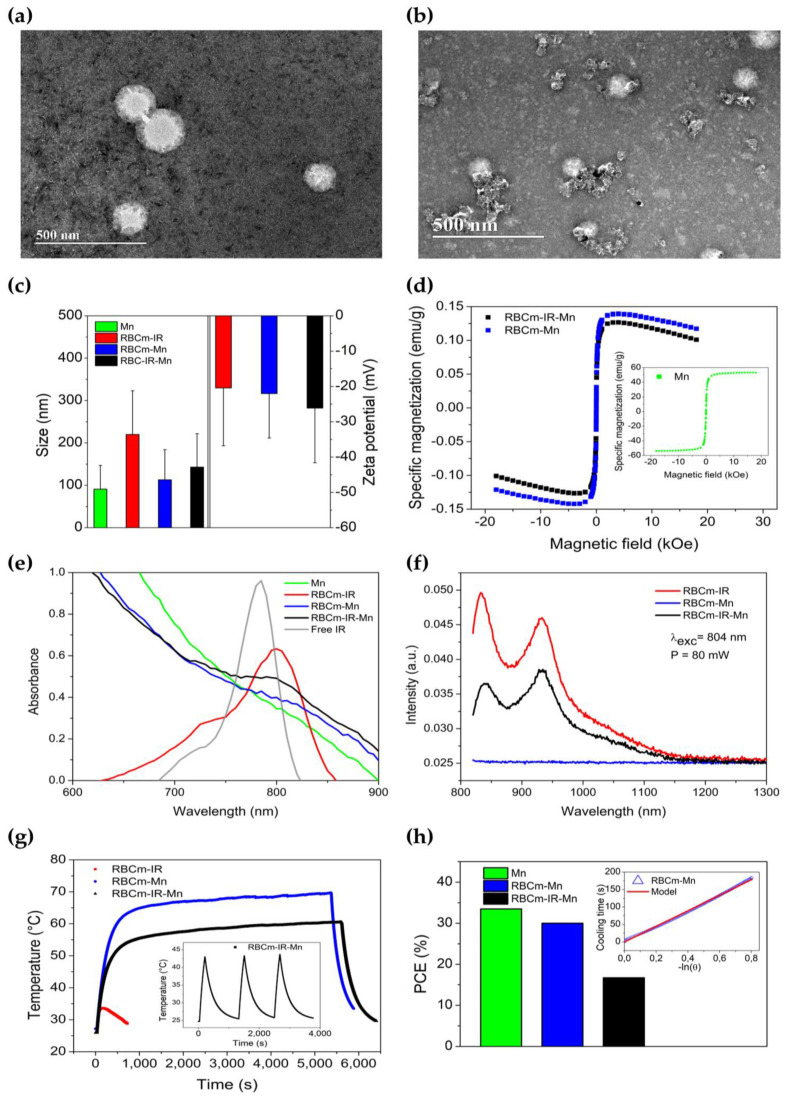
RBCm-IR-Mn nanocarrier characterization. TEM images of (**a**) RBCm-IR and (**b**) RBCm-IR-Mn. (**c**) Mean diameters (obtained by NTA) and zeta potentials for RBCm-IR, RBCm-Mn, and RBCm-IR-Mn. Mean size of Mn-NP aggregates also included. (**d**) VSM measurements for RBCm-Mn and RBCm-IR-Mn. Inset: measurements for Mn-NPs (powder). (**e**) Absorption spectra of RBCm-IR-Mn and its components (samples measured at different concentrations). (**f**) RBCm-IR-Mn, RBCm-IR, and RBCm-Mn fluorescence spectra (80 mW, 804 nm). (**g**) RBCm-IR, RBCm-Mn, and RBCm-IR-Mn PTT profiles (965 mW, 808 nm). Inset: RBCm-IR-Mn photostability study. (**h**) Mn NP, RBCm-Mn, and RBCm-IR-Mn photothermal conversion efficiencies (PCE, not calculated for RBCm-IR due to photobleaching). Inset: best fit of Roper´s model (for the cooling regime, laser off) to the experimental data for RBCm-Mn.

**Figure 2 pharmaceutics-15-00943-f002:**
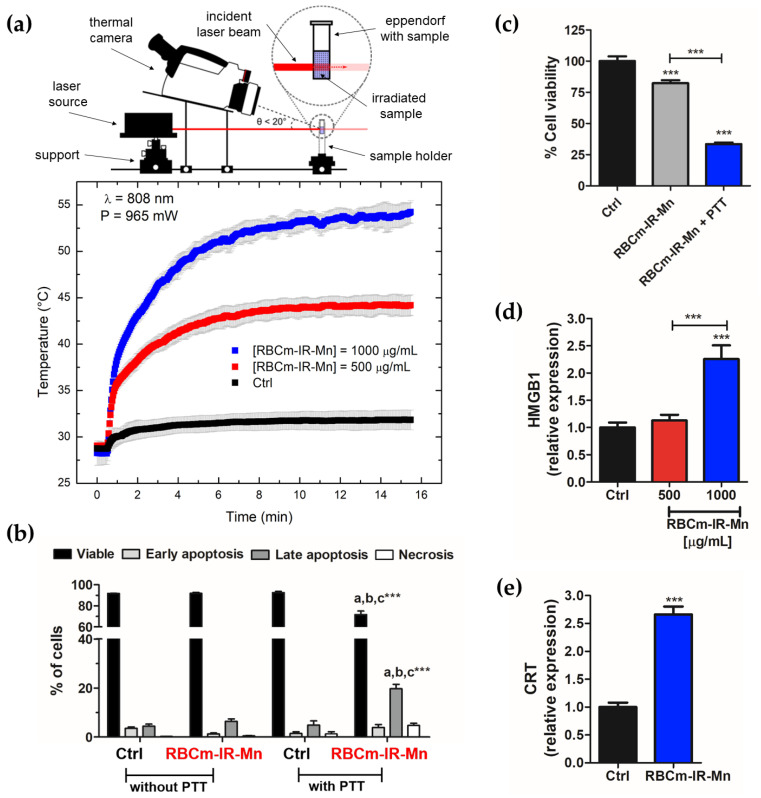
Photothermal effect on cell viability and cell death mechanism. (**a**) PTT (965 mW, 808 nm) profile of S180 cells containing RBCm-IR-Mn at distinct particle concentrations, 500 and 1000 µg/mL. Control: S180 cells without the nanocarrier. Datapoints show the mean value, while error bars represent standard deviation. (**b**) Flow cytometry analysis quantifying the death mechanism for S180 cells treated with RBCm-IR-Mn at 500 µg/mL, without and with PTT. Control: S180 cells not treated with the nanocarrier. The figure shows the percentage of either viable cells, or of cells in early apoptosis, late apoptosis, or necrosis. (**c**) MTT study of the S180 cells with RBCm-IR-Mn at 500 µg/mL, without and with PTT. Control: S180 cells submitted to PTT. (**d**) HMGB1 expression from S180 cells 24 h after RBCm-IR-Mn-mediated PTT, at distinct particle concentrations, 500 and 1000 µg/mL. (**e**) Calreticulin (CRT) expression from S180 cells 24 h after PTT mediated by RBCm-IR-Mn at 1000 µg/mL. Letters a, b, and c refer to Ctrl without PTT, RBCm-IR-Mn without PTT, and Ctrl with PTT, respectively. *** *p* < 0.001.

**Figure 3 pharmaceutics-15-00943-f003:**
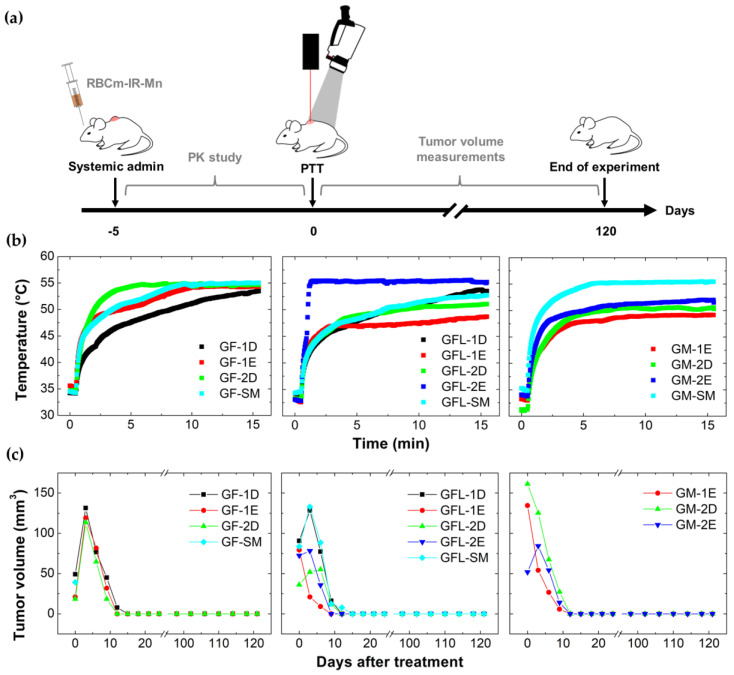
In vivo preclinical PTT study. (**a**) Schematic image of the PTT experimental design. RBCm-IR-Mn is intravenously administered on Day-5, PTT happens on Day 0, and tumor growth is monitored for up to 120 days. (**b**) PTT profiles for animals of groups GF, GFL, and GM. Each curve corresponds to one animal. (**c**) Tumor growth evolution after PTT for groups GF, GFL, and GM. Animal GM-SM is not shown (see Appendix A). Eleven animals showed complete tumor regression after RBCm-IR-Mn-mediated PTT. Average PTT and tumor growth profiles are provided in Appendix A.

**Figure 4 pharmaceutics-15-00943-f004:**
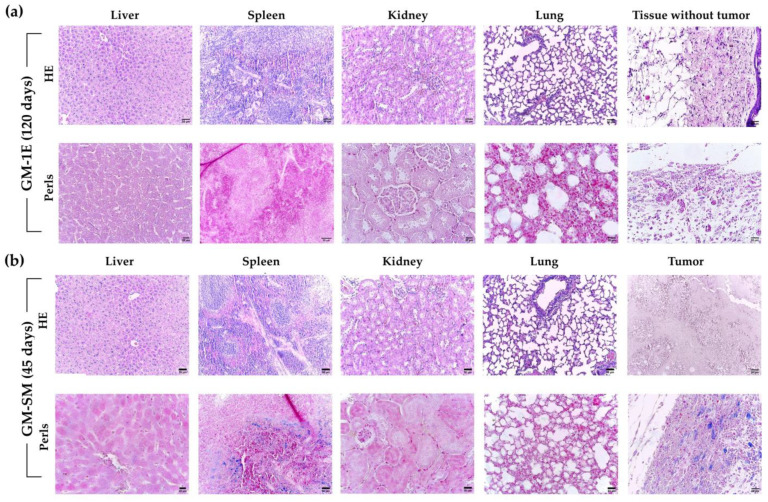
Histopathological analysis. (**a**) Animal GM-1E (evaluated 120 days after PTT). (**b**) Animal GM-SM (evaluated 45 days after first PTT session). Hematoxylin and eosin (HE) and Perls (iron identification) staining in tissues and tumor. No tissue damage in liver, spleen, kidney, and liver of GM-1E and GM-SM (HE). GM-1E HE evidenced complete absence of tumor tissue. GM-SM HE presented characteristic tumor tissue, with several areas of necrosis/apoptosis. Perls indicated moderate presence of iron only in the spleen, and an intense amount of iron in the tumor of GM-SM (blue spots). Scale bars equate 50 μm, except for GM-1E Perls in spleen and GM-SM Perls in liver, where they equate 20 μm.

**Figure 5 pharmaceutics-15-00943-f005:**
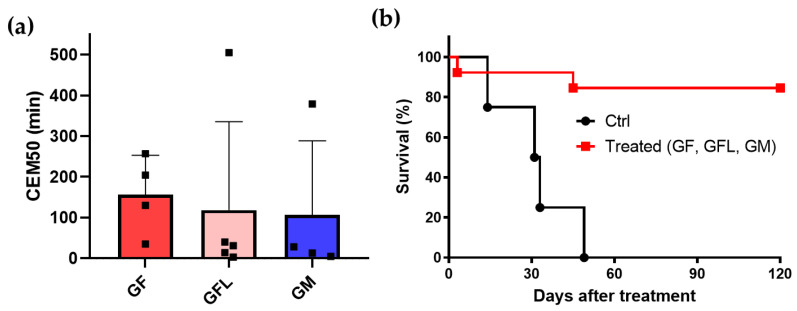
In vivo PTT results. (**a**) Thermal dose (CEM50) for groups GF, GFL, and GM, submitted to PTT under ablative regime. (**b**) Survival (Kaplan–Meyer) curve of the control (nontreated) and PTT-treated animals (groups GF, GFL, and GM together). The overall survival rate, for animals monitored for up to 120 days, was 85% (11/13).

## Data Availability

Not applicable.

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
