# Peer review of "Immunogenic Cell Death Photothermally Mediated by Erythrocyte Membrane-Coated Magnetofluorescent Nanocarriers Improves Survival in Sarcoma Model"

_pharmaceutics, 2023, doi:10.3390/pharmaceutics15030943_

Round 1
Reviewer 1 Report
The present draft appears to me to be of excellent quality in terms of content and form. By way of qualification, I would like to state that, as a physicist and materials scientist, I am referring primarily to the first part. The mouse model experiments carried out and their evaluation also appear to me to be excellently designed and evaluated, although my own expertise does not permit a final evaluation here.
I recommend publication of the present draft, but still suggest some adaptation and comments:
(1) The present paper makes only partial statements about the structural makeup of the nanoparticles presented here, and this should perhaps be explicitly clarified again. The nanoparticles are characterized with respect to very essential properties necessary to perform the following experiments, in this respect the presentation is complete, however, the reader is left with some unanswered questions:
(a) Why are the RBCm-IR significantly larger than the RBCm-IR-Mn particles? This is probably due to the manufacturing process? How does this size difference affect subsequent experiments?
(b) The absorption spectrum of the RBCm-IR-Mn particles is apparently dominated by the Mn. Could it be that the IR-780 density in the RBCm-IR and RBCm-IR-Mn particles is simply not comparable?
(c) What is the cause of the red-shift and quenching? How is this related to the structural makeup of the RBCm-IR-Mn particles?
(d) Could the effect of strong reduction of photobleaching, which is essential for the following experiments, be related to the (assumed) structural difference of the RBCm-IR and RBCm-IR-Mn particles?
If the authors have data on this, I recommend including them in the publication. If these data are not available (which I assume they are) I recommend at least commenting on them accordingly.
(2) I would also like to add the following comments:
(a) Figure 1(e) uses blue color twice, which makes it difficult to distinguish IR from RBCm-Mn.
(b) Figure 1(h) does not contain any error bars.
(c) Line 450 refers to Figure 5c, more likely Figure 2d is meant here.
(d) The measurement setup in Figure 2(a) is not self-explanatory and relatively small (especially the lower black plots).
Overall, I have read the publication with great pleasure and interest. Congratulations on your work!
Reviewer 2 Report
This study put a few functional components together to build a medicine for sarcoma treatment. The research is within the scope of this journal and should be useful for plenty of readers. There are some issues that needs to be improved for publication though.
1. Please rephrase the title, as the current title is not relevant enough to the main data and experimental content of the article.
2. Please add the necessary control groups and select a larger range of concentration gradients to support the results.
3. Figure1(b) is not clear enough to support the claims of the article.
4. The data in Figure3 can be further refined and it is necessary to refer to others. A larger sample size would make the results more meaningful.
5. Please better incorporate the three components of the preparation into the content of the experiment and discuss, and the language description should focus on the significance of the composition of reagents.
Reviewer 3 Report
It is an interesting paper. Authors evaluated in detail the in vitro and in vivo performance of RBCm-IR-Mn nanocarriers for photothermal therapy applications in the ablation regime using the murine sarcoma model S180 in Swiss albino mice. Several characterization techniques are used to investigate the biomedical potential of this nanocarrier. o evaluated how the tumor heterogeneity influences the thermal dose delivery at constant laser power condition in the ablation regime, as well as if the intratumoral heat generation during PTT is influenced by tumor matrix. The role of animal gender on the heat delivery was also addressed. Finally, the tumor growth profiles after PTT mediated by systemic administration of RBCm nanocarriers were moni-tored up to 120 days to evaluate the possibility of tumor recurrence. We present strong evidence that PTT, at the ablative regime, induced immunological response resulting in a very high survival rate 85% in the murine S180 model. It can be accepted as is.
Author Response
We are grateful to the reviewer for generously dedicating time to review our work and for kindly providing this very positive feedback.
Round 2
Reviewer 2 Report
I have no further questions. Though many issues are not directly solved, I could understand the limit of the authors.